# Identifying graft incompatible rootstocks for sweet cherry through machine learning algorithms

Erol Aydın[1]*, Mehmet Ali Cengiz[2], Ercan Er[1], Hüsnü Demirsoy[3]

1 Department of Horticulture, Black Sea Agricultural Research Institute, Gelemen, Samsun, Türkiye,
2 Department of Mathematics and Statistics, College of Science, Imam Mohammad Ibn Saud Islamic University (IMSIU), Riyadh, Saudi Arabia, 3 Department of Horticulture, Faculty of Agriculture, Ondokuz Mayıs University, Atakum, Samsun, Türkiye

* aydin.erol@tarimorman.gov.tr

## Abstract

Graft incompatibility is a key factor in the development of dwarf and semi dwarf rootstocks for sweet cherry (*Prunus avium* L.) to improve yield, fruit quality, precocity, and labor efficiency. This study evaluated the graft incompatibility of eight genotypes three sweet cherry, three sour cherry, and two mahaleb collected from Northern Anatolia, a native region for cherries. These genotypes, along with standard rootstocks Gisela 6 and SL 64, were grafted with '0900 Ziraat' and 'Lambert' cultivars. Graft incompatibility was assessed using a multidisciplinary approach combining classical morphological and anatomical evaluations with advanced data driven analyses. Parameters such as graft bud growth rate (40.26–86.21%), shoot length (41.01–91.28 cm), and rootstock/scion diameter ratio (0.41–0.92) were measured 12 months after grafting. Principal Component Analysis, Random Forest modeling with SHAP values, and Bayesian ranking were applied to identify key traits and rank genotype performance. The integrated analysis successfully distinguished compatible rootstock candidates, identifying five genotypes with high compatibility potential. These findings demonstrate that combining traditional phenotypic evaluation methods with machine learning-based approaches offers a robust and comprehensive framework for addressing graft incompatibility, and contributes valuable insights for future breeding programs and rootstock selection strategies in sweet cherry.

## 1. Introduction

The sweet cherry (*Prunus avium* L.) is native to certain parts of Northern Anatolia [1,2]. The main diversity centers of both sweet (*Prunus avium*) and sour (*Prunus cerasus*) cherries are in Asia Minor, northern Iran, Iraq, Syria, Ukraine, and countries south of the Caucasus Mountains [3]. Theophrastus first mentioned cherries around 300 BC as Kerasos, and the Greek word Kerasia likely derives from this. The species

**Data availability statement:** The datasets used and analyzed during the current study are publicly available on Zenodo at the following DOI: https://doi.org/10.5281/zenodo.16944180.

**Funding:** This work was supported and funded by the Deanship of Scientific Research at Imam Mohammad Ibn Saud Islamic University (IMSIU) (grant number IMSIU-DDRSP2502).

**Competing interests:** The authors declare that there are no conflicts of interest.

name cerasus is believed to have originated from the city of Giresun in the Black Sea region, known historically as Kerasia, which still harbors great cherry diversity [4].

As in many fruit species, rootstocks are vital in sweet cherry cultivation. They affect propagation success, climate and soil adaptability, disease and pest resistance, yield, fruit quality, and tree vigor. Globally, key rootstocks include seedling or clonal types of *Prunus avium* and *Prunus mahaleb*. Notably, wild *P. avium* types known as 'mazzard', 'wild cherry', or 'gean' have been used since 330–400 B.C. by Greeks and Romans [5].

French pomologists began using *P. mahaleb* ('St Lucie' or 'perfume cherry') as a rootstock in 1768. Mazzard cherry (*P. avium*) seedlings had already been used by ancient Greek and Roman gardeners about 2400 years ago. In the last 75 years, efforts to improve limitations of mazzard rootstocks led to the development of clonally propagated types, starting with F.12/1. In the late 1950s, the East Malling Research Station (UK) introduced Colt (*P. avium* x *P. pseudocerasus*), the first semi dwarf rootstock [6]. However, in many studies and production trials, it has been observed that the Colt rootstock is not truly semi dwarf, as it tends to produce larger trees. Germany's Geissen program, launched in 1965, produced the Gisela series in the 1980s a major breakthrough. Gisela 5 and Gisela 6 gained global popularity for their dwarfing ability and high productivity [7]. More recently, hybrid clonal series like P-HL, Pi-Ku, Camil (GM 79), Damil (GM 61/1), Inmil (GM 9), and MaxMa (M×M) have also been developed [8].

Graft incompatibility is a major concern in propagation by grafting. While grafting offers substantial horticultural advantages, the biological mechanisms underlying scion-rootstock compatibility remain unclear [9]. In sweet cherries, especially when grafted onto *P. mahaleb*, incompatibility is common and may lead to plant decline or death over time. It is defined as the failure of graft partners to form a functional, unified tissue, followed by physiological attempts to integrate as one plant. Graft incompatibility is categorized as either translocated or localized [10,11]. The former manifests within the first year as growth cessation, defoliation, and leaf discoloration, while the latter frequently seen in sweet cherry emerges later due to biochemical alterations at the graft union. This delayed incompatibility causes anatomical anomalies, vascular disruption, necrosis, and eventual graft failure [12–16]. Such issues, particularly in cherry/mahaleb combinations, can result in significant long-term economic losses.

*P. avium* rootstocks generally exhibit good compatibility with sweet cherry cultivars. However, most modern rootstocks are hybrids involving species such as *P. cerasus, P. canescens, P. pseudocerasus,* and *P. mahaleb*, with their genetic diversity offering desirable traits for growers. Yet, this same diversity can sometimes lead to graft incompatibility with specific cherry cultivars. Moreover, such incompatibility may become more pronounced under stress conditions [17].

*Prunus mahaleb* gained popularity in the U.S. in the mid-19th century and became the dominant sweet cherry rootstock by the early 1900s due to its ease of seed propagation and tolerance to cold and some diseases. However, by the 1920s, incompatibility with many cultivars was noted, leading to early tree mortality. Recently,

incompatibility symptoms have been reported with Lapins, Chelan, and Tieton in the Pacific Northwest. In Türkiye's Northern Central Anatolia, where soils are calcareous, mahaleb remains widely used [6]. With increasing climate challenges, its drought and lime tolerance may enhance its future value [18].

Given the need for rootstocks that ensure compatibility, precocity, and enhanced yield and fruit quality, the Black Sea Agricultural Research Institute initiated a program in 2009 to collect cherry, sour cherry, and mahaleb genotypes from Giresun the native region of cherries. Numerous wild, sweet, and sour cherry genotypes were evaluated, focusing on their growth and vigor. Among these, eight genotypes (three sweet cherry, three sour cherry, and two mahaleb) showed promise as rootstock candidates. This study aimed to evaluate their graft compatibility with the sweet cherry cultivars '0900 Ziraat' and 'Lambert'.

In recent studies of genotype evaluation and graft incompatibility, classical statistical techniques such as ANOVA, correlation analysis, and basic scoring systems have frequently been employed to assess phenotypic traits and physiological responses. While these methods provide valuable descriptive insights, they often fall short in capturing complex multivariate relationships and in delivering robust, generalizable conclusions especially when high-dimensional and nonlinear interactions are involved. To overcome these limitations, this study integrates Machine learning techinuques namely, Principal Component Analysis (PCA) with unsupervised clustering, Random Forest combined with SHAP values, and a Bayesian ranking model to enhance the interpretability, prediction, and reliability of genotype evaluations.

While machine learning (ML) methods have shown growing promise in plant science, their specific application to graft compatibility remains relatively unexplored. Recent studies have demonstrated the potential of ML in fruit related predictions and physiological trait modeling. For instance [19], Yang et al. (2024) applied deep neural networks to predict morphological characteristics in grafted *Camellia oleifera* plants, while [20] Sabouri et al. (2025) utilized ML techniques for non-destructive estimation of plum fruit weight [21]. Cadet et al. (2019) combined infrared spectral data with ML models to characterize apricot fruit firmness, indicating a role for data driven approaches in fruit texture analysis. Additionally [22], Yang et al. (2023) emphasized the use of ML in understanding physiological responses under stress conditions, highlighting its broader applicability in plant functional studies. Despite these advances, previous works largely focus on predictive performance without prioritizing model interpretability. In contrast, the present study introduces a hybrid framework that integrates principal component analysis (PCA) for dimensionality reduction, random forest for nonlinear prediction, SHAP values for model transparency, and Bayesian ranking for trait prioritization. This integrative and explainable framework offers new insights into the anatomical and physiological predictors of graft compatibility in sweet cherry genotypes.

Moreover, Bayesian regression and ranking provide probabilistic assessments of genotype performance, accounting for uncertainty and allowing direct inference on the relative superiority of each genotype with credible intervals and posterior probabilities.

This study aims to evaluate the graft compatibility of eight promising cherry rootstock candidates collected from the native cherry region of Giresun with the sweet cherry cultivars '0900 Ziraat' and 'Lambert'. Recognizing the importance of rootstock compatibility, vigor, and adaptability, the research focuses on wild sweet cherry, sour cherry, and *Prunus mahaleb* genotypes. To overcome the limitations of traditional evaluation methods, advanced data driven approaches such as PCA with clustering, Random Forest with SHAP, and Bayesian ranking were employed. These methods enhance interpretability and prediction, enabling a more robust assessment of genotype performance under complex trait interactions.

## 2. Materials and methods

### 2.1 Material

In this study, two sweet cherry and three sour cherry genotypes previously characterized by UPOV criteria and collected from Kerasi (Giresun) and surrounding areas were evaluated. These genotypes, identified as dwarf or semi dwarf with promising potential for tissue culture and propagation [23] Koç & Bilgener (2013), were selected for further assessment as

rootstocks. Their graft compatibility was compared with standard semi vigorous to vigorous rootstocks Gisela 6 and SL 64. The sweet cherry cultivars 'Lambert' (upright and vigorous) and '0900 Ziraat' (vigorous and upright spreading) were used as scions in the experiment.

## 2.2  Methods

All experimental data were collected from trees grown at the Black Sea Agricultural Research Institute (Gelemen, Samsun, Turkey) with prior permission granted by the Institute's administration. In 2019–2020, '0900 Ziraat' and 'Lambert' sweet cherry varieties were grafted with bud grafts on sweet cherry rootstock candidate genotypes and cherry rootstocks propagated by *in vitro* propagation in mid-August. Anatomical examinations were performed 12 months after grafting on variety/rootstock combinations. All Cultivars/rootstock combinations were made on 60 grafts (A total of three replicates were conducted, each consisting of 20 grafted).

For anatomical analysis, 10 cm long graft samples (n = 5) were fixed in FAA solution for 24 hours, then transferred to 70% alcohol. Tissues were softened in a 1:1 glycerine alcohol mixture before sectioning. Transverse (30–50 µm) and longitudinal sections were obtained from the graft union using a slide microtome, following the protocol by [24]. Sections were stained with 1% iodinated potassium iodide (IKI), dried with blotting paper, and mounted in melted glycerine gelatine. To detect starch accumulation, longitudinal sections were also stained with 1% potassium iodide, and visual assessment focused on combinations showing the highest accumulation in the scion [15]. For each grafted genotype scion combination, five replicates were used (n = 5). Although measurements were taken from all replicates individually, statistical analyses were performed based on the average values of these five replicates to represent each combination.

Previous studies have shown that phenolic compounds, starch, certain enzymes, and proteins in scion and rootstock tissues influence graft incompatibility [9,10,12,14,24–26]. The accumulation of these substances around the graft union affects callus formation, tissue differentiation, and necrotic layer development. Based on this knowledge, microscopic evaluations in the present study focused on callus formation, cambial differentiation, necrosis, and cambial continuity at the graft interface. Callus formation was scored on a 1–10 scale, with 10 indicating the most vigorous callus and 1 the least. Necrotic layer formation was similarly rated, where 1 indicated severe necrosis and 10 minimal necrosis. Cambial continuity was assessed by the degree of interruption in the cambial tissue, scoring 10 for uninterrupted continuity and 1 for the most disrupted combinations.

### 2.2.1  Growth characteristics.
Numerous studies have identified graft bud growth rate (%), graft shoot length, rootstock/scion diameter ratio, defoliation time, leaf discoloration timing, and starch accumulation as key indicators of graft incompatibility [5,6,15,27]. Accordingly, these parameters were assessed in this study. Graft bud growth rate was calculated as the percentage of shoots emerging from the graft union. Shoot length was measured one-year post-grafting at the onset of dormancy using a measuring tape. The rootstock/scion diameter ratio was determined by dividing the diameter measured 5 cm below the graft union by the diameter 5 cm above it, using a digital calliper. Defoliation was scored from 10 (earliest) to 1 (latest), based on the assumption that early leaf fall reduces cold damage. Leaf discoloration was visually scored from 1 (earliest discoloration) to 10 (latest). For starch accumulation, combinations were rated from 1 (highest accumulation) to 10 (lowest) based on visual assessment at the graft site.

### 2.2.2  Statistical analysis.
Given the multifactorial nature of graft compatibility, relying solely on classical descriptive statistics may be insufficient to capture the complex interrelationships among phenotypic traits. Therefore, in this study, a combination of advanced statistical and machine learning techniques was employed to analyse and interpret the data with greater precision. These approaches not only enhance the understanding of trait interactions but also provide a robust foundation for identifying high-performing genotypes. The following subsections outline the analytical framework used to reduce dimensionality, predict compatibility outcomes, and rank genotypes probabilistically (Fig 1).

Multivariate Trait Reduction and Genotypic Clustering via PCA and K-Means: To explore the intrinsic structure of the multivariate dataset and to identify natural groupings among genotypes based on trait profiles, Principal Component

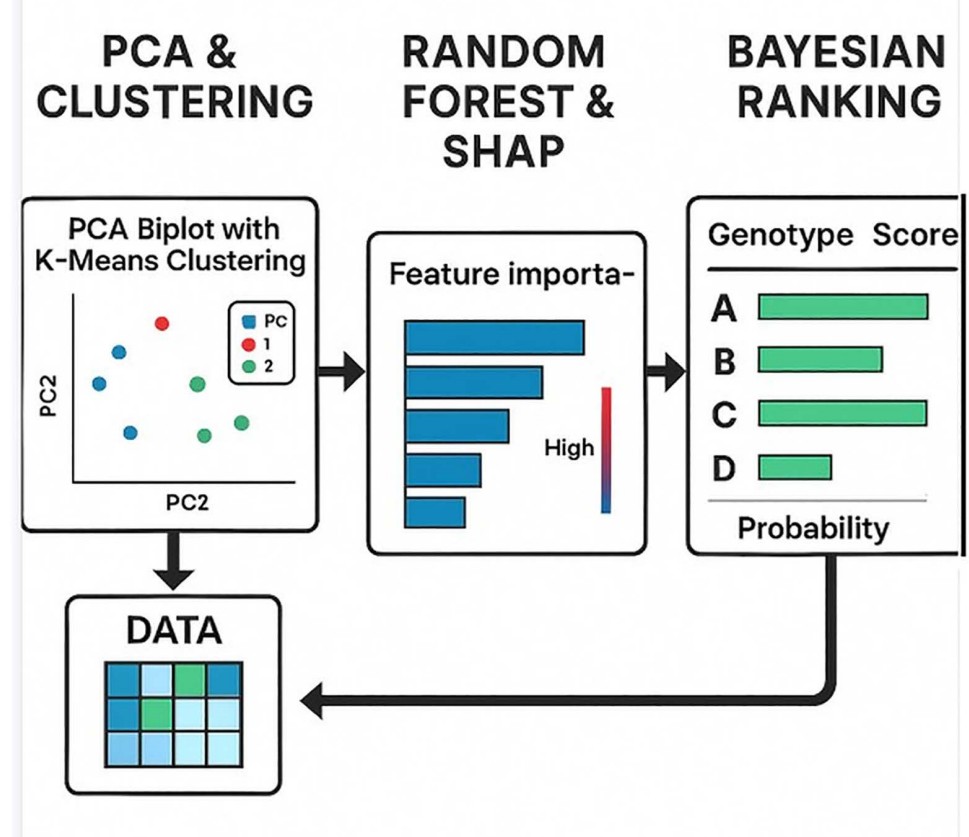

**Fig 1. Overview of the integrated hybrid analysis framework applied in this study.**

Analysis (PCA) was employed for dimensionality reduction [28]. All trait variables were first normalized to the [0.1] scale to ensure comparability. The first two principal components were retained to capture the majority of variance. Subsequently, K-Means clustering was applied on the PCA scores to group genotypes into homogeneous clusters [29]. This unsupervised learning approach allowed us to detect underlying similarities in genotype responses without prior labelling, offering an interpretable two-dimensional visualization of complex relationships.

Predictive Modelling and Feature Importance via Random Forest and SHAP: To quantify the predictive contribution of each trait to overall graft compatibility, a Random Forest regressor was trained using normalized trait data to predict a composite compatibility score [30]. The model's performance was assessed using internal validation metrics (e.g., $R^2$, MAE). To enhance interpretability, SHAP (SHapley Additive exPlanations) values were computed [31]. These provide both global feature importance rankings and local explanations for individual predictions, highlighting which traits increase or decrease the compatibility score across genotypes. This approach offers model-agnostic, human-interpretable insights into the complex interactions among traits.

Probabilistic Genotypic Ranking via Bayesian Regression: To incorporate uncertainty and to provide a probabilistic ranking of genotypes, a Bayesian linear regression model was constructed [32]. Trait variables were treated as covariates and the composite compatibility score was standardized prior to modelling. Posterior distributions of the model coefficients were obtained via Markov Chain Monte Carlo (MCMC) sampling using the PyMC framework [33]. Based on the posterior predictive distribution, each genotype was assigned a latent compatibility score ($\theta_i$), from which both 95% credible

intervals and probabilities of being the best-performing genotype were derived. This probabilistic framework allows for direct comparison among genotypes while accounting for uncertainty in parameter estimates.

The pipeline begins with PCA and K-means clustering, which are used to reduce the dimensionality of the multivariate trait dataset and to identify natural groupings among genotypes based on shared patterns. In the second stage, Random Forest modelling is performed to predict compatibility scores, and SHAP values are used to quantify the relative importance of each trait in model predictions. Finally, a Bayesian regression model is used to calculate latent compatibility scores ($\theta$) for each genotype, allowing posterior ranking and estimation of the probability of being the best-performing genotype. This integrative workflow enhances both interpretability and inference in genotype evaluation under graft compatibility assessments.

Table 1 outlines the objective, typical application scenarios, and specific contributions of each method (PCA+clustering, Random Forest with SHAP, and Bayesian regression) to the evaluation of graft compatibility among genotypes.

**2.2.3 Machine learning parameters.** The Random Forest model was implemented using the Random Forest Regressor class from the Scikit learn library. The number of trees in the forest was set to 500 (n_estimators=500), and the random state was fixed at 42 to ensure reproducibility. Default settings were used for all other hyper parameters, including max depth, min samples split, and max features. The model was trained on the full scaled dataset, and predictions were generated for performance evaluation and feature importance analysis.

K-means clustering was applied with k=3 clusters, chosen based on prior knowledge of graft compatibility groupings. While no formal metric was used to optimize cluster number, this choice reflects known biological groupings among genotypes.

The Bayesian ranking model was implemented in PyMC using a hierarchical linear regression structure. Normal priors ($\mu=0$, $\sigma=1$) were assigned to both the intercept and the regression coefficients, while a half normal ($\sigma=1$) prior was used for the residual standard deviation. The model was sampled using the No-U-Turn Sampler (NUTS) with 4 independent chains. Each chain included 1.000 tuning steps and 2.000 sampling iterations, resulting in 8.000 total posterior draws. The target acceptance probability was set to 0.95, and a fixed random (42) ensured reproducibility. Convergence diagnostics were assessed using the Gelman Rubin statistic ($\hat{R}$), with all parameters achieving values below 1.01, indicating proper convergence and mixing. Posterior samples of the intercept and coefficients were combined to compute genotype level posterior scores ($\theta$). For each genotype, the posterior mean, 95% Highest Density Interval (HDI), mean rank across draws, and the probability of being ranked as the top-performing genotype were calculated. These probabilistic outputs provide an interpretable and uncertainty-aware ranking of graft combinations.

## 3. Results

### 3.1 Cultivars/rootstock combinations occurred with the '0900 Ziraat' cultivar

Among all rootstock combinations with the '0900 Ziraat' cultivar, 52 M 003 genotype showed the lowest callus formation and cambium continuity, while Gisela 6 rootstock exhibited the highest performance in both traits (Table 2). Cambium differentiation was lowest in 28 M 005 genotype and highest in Gisela 6 rootstock. Regarding necrotic layer formation, 28

**Table 1. Overview of the analytical methods used in this study.**

| Analysis Type | Purpose | When to Use | Contribution in this study |
|---|---|---|---|
| PCA+Clustering | Detects general patterns in the data and separates genotypes into clusters | To explore structural similarities, reduce dimensionality and visualize multivariate traits | Which genotypes are similar? Do compatible clusters naturally emerge? |
| Random Forest+SHAP | Predicts scores and identifies how important each trait is | To explain model behavior, interpret variable importance and uncover casual relationships | Which traits are most influential? SHAP enables intuitive, model aware explanation |
| Bayesian Model | Generates probabilistic compatibility scores and ranks genotypes | To rank with uncertainty and generalize classical scoring models | What is the probability that Gisela 6 is the best? How uncertain is the ranking? |

**Table 2. Data of the measurement related to graft incompatibility of the '0900 Ziraat' cultivar grafted onto different rootstock candidates.**

| Characteristic | Gisela 6 | 08 K 056 | 52 K 063 | 55 K 104 | 28 V 001 | 28 V 003 | 55 V004 | 52 M 003 | 28 M 005 | SL 64 |
|---|---|---|---|---|---|---|---|---|---|---|
| Ability to form callus* | 10.0 | 8.33 | 6.77 | 8.67 | 7.33 | 7.33 | 7.67 | 5.0 | 6.33 | 8.3 |
| Cambium differentiation* | 9.0 | 7.67 | 6.33 | 8.0 | 7.0 | 7.33 | 7.0 | 4.67 | 4.0 | 6.8 |
| Necrotic layer formation* | 9.0 | 8.0 | 6.0 | 7.66 | 7.67 | 7.0 | 7.33 | 6.67 | 5.0 | 7.5 |
| Cambium continuity* | 10.0 | 8.33 | 6.77 | 8.0 | 7.67 | 7.33 | 7.67 | 5.0 | 6.67 | 8.0 |
| Graft bud growth rate (%) | 86.21 | 72.17 | 67.42 | 77.18 | 74.11 | 75.37 | 71.54 | 57.29 | 47.81 | 79.54 |
| Length of graft shoot (cm) | 77.36 | 64.08 | 52.61 | 91.28 | 65.42 | 72.83 | 66.29 | 41.01 | 47.86 | 92.38 |
| Rootstock/scion diameter | 0.88 | 0.80 | 0.62 | 0.64 | 0.66 | 0.70 | 0.72 | 0.52 | 0.41 | 0.76 |
| Defoliation time* | 8.0 | 9.0 | 6.77 | 7.67 | 7.33 | 7.0 | 7.33 | 6.33 | 5.0 | 7.0 |
| Time of leaf discoloration* | 9.0 | 8.33 | 7.33 | 8.0 | 7.67 | 7.33 | 7.0 | 6.0 | 6.67 | 8.0 |
| Starch accumulation* | 10.0 | 7.33 | 6.33 | 7.67 | 7.67 | 7.33 | 7.67 | 5.0 | 5.67 | 8.3 |

*(1–10) Visual evaluation was made on a 1–10 scale.

M 005 genotype had the least, and Gisela 6 rootstock the most. Graft bud growth rate ranged from 47.81% (28 M 005) to 86.21% (Gisela 6), and shoot length varied from 41.01 cm (52 M 003) to 92.38 cm (SL 64). The lowest rootstock/scion diameter ratio was in 28 M 005 (0.41) genotype, and the highest in Gisela 6 (0.88) rootstock. Defoliation occurred earliest in 28 M 005 genotype and latest in 08 K 056 genotype. Leaf discoloration was most intense in 52 M 003 and least in Gisela 6. Similarly, the highest starch accumulation at the graft union was observed in 52 M 003 genotype, and the lowest in Gisela 6 rootstock (Figs 2 and 3).

As visually demonstrated in Fig 2. notable graft bud development was observed in compatible combinations. This visual evidence aligns with the quantitative measurements of callus formation and cambial continuity shown in Table 2.

### 3.2 Cultivars/rootstock combinations occurred with the 'Lambert' cultivar

In the 'Lambert' sweet cherry cultivar, callus formation was lowest in 28 M 005 genotype and highest in Gisela 6 rootstock (Table 3). Cambium differentiation was weakest in 52 M 003 genotype and strongest in Gisela 6 rootstock. Necrotic layer formation was minimal in genotype 28 M 005 and maximal in Gisela 6 rootstock. Cambium continuity was highest in Gisela 6 and lowest in 52 M 003 genotype, similar to findings with '0900 Ziraat'. Graft bud growth rate ranged from 40.26% (28 M 005) to 81.08% (Gisela 6), while shoot length varied from 43.62 cm (52 M 003) to 89.39 cm (SL 64). The rootstock/scion diameter ratio was lowest in 28 M 005 (0.47) genotype and highest in Gisela 6 (0.92) rootstock. Defoliation occurred earliest in 28 M 005 and latest in 08 K 056 genotype. Leaf discoloration was most pronounced in 52 M 003 genotype and least in Gisela 6 rootstock. Starch accumulation at the graft site followed the same trend, with the highest in 52 M 003 genotype and the lowest in Gisela 6 rootstock.

Fig 4. visualizes the distribution of genotypes in the reduced trait space defined by the first two principal components (PC1 and PC2), which together capture the majority of the variance in the dataset. Genotypes are grouped using the K-Means clustering algorithm applied on the PCA transformed data. Each point represents a genotype, and colors indicate cluster membership: Cluster 0 (red), Cluster 1 (blue), and Cluster 2 (green). Notably, Gisela 6 forms a distinct cluster (green), indicating a unique trait pattern. In contrast, 28 M 005 and 52 M 003, which appear isolated in Cluster 1, are associated with weaker graft compatibility. This unsupervised clustering provides early evidence of genotypic divergence and supports subsequent supervised and Bayesian analyses.

The PCA biplot presented in Fig 4. illustrates how the genotypes distribute within the two-dimensional trait space defined by the first two principal components (PC1 and PC2). These components, derived from the normalized phenotypic traits, capture the major variation in the dataset and enable an interpretable visualization of the multivariate structure.

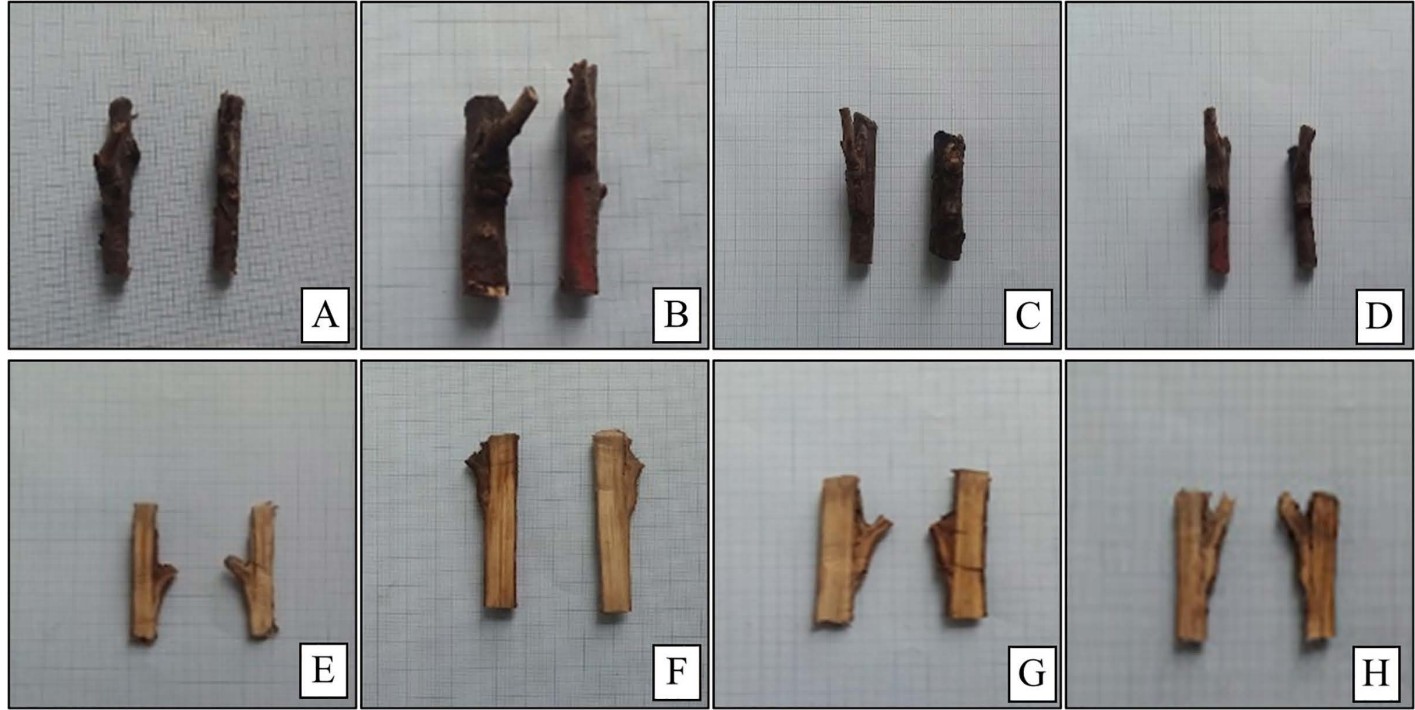

**Fig 2. Grafting development on variety/rootstock combinations after 12 months. (A and E):08 K 056/0900 Ziraat, (B and F):55 K 104/Lambert, (C and G): 55 V 004/0900 Ziraat, (D and H): Gisela 6/0900 Ziraat.**

The application of K-Means clustering on the PCA scores revealed three distinct genotype clusters. The majority of genotypes, including SL 64, 55 K 104, 28 V 003 and others, were grouped in Cluster 0 (red), suggesting a shared phenotypic pattern and potentially moderate levels of graft compatibility. In contrast, Cluster 1 (blue) contained only a few genotypes such as 28 M 005, 52 M 003, and 52 K 063 that appeared more isolated in the PCA space, indicating divergent trait profiles. These genotypes were later confirmed to have lower compatibility scores in the subsequent Random Forest and Bayesian analyses, supporting the unsupervised findings.

Most notably, Gisela 6 emerged as the sole member of Cluster 2 (green), clearly separated from all other genotypes along the first principal component. This suggests that Gisela 6 exhibits a unique combination of traits that sets it apart phenotypically. The distinct position of Gisela 6 in PCA space is consistent with its high compatibility ranking in the Bayesian posterior model, where it was identified with over 95% probability as the best-performing genotype.

Overall, this PCA based clustering provides an unsupervised validation of genotypic differentiation and offers an early glimpse into the latent structure of compatibility-related traits. The findings from this figure not only support the predictive models used later in the study but also highlight the utility of dimensionality reduction and cluster analysis in biological genotype evaluation.

The relative contribution of each phenotypic trait to the model's predictive performance. Higher values indicate greater influence on the predicted compatibility score. Among all traits, graft bud growth rate, cambium continuity and ability to form callus were identified as the most influential predictors, while time of leaf discoloration contributed the least to model output (Fig 5).

The Random Forest model provided an interpretable ranking of trait importance based on their predictive power for graft compatibility. As shown in Fig 5. graft bud growth rate emerged as the most important trait, indicating that faster bud

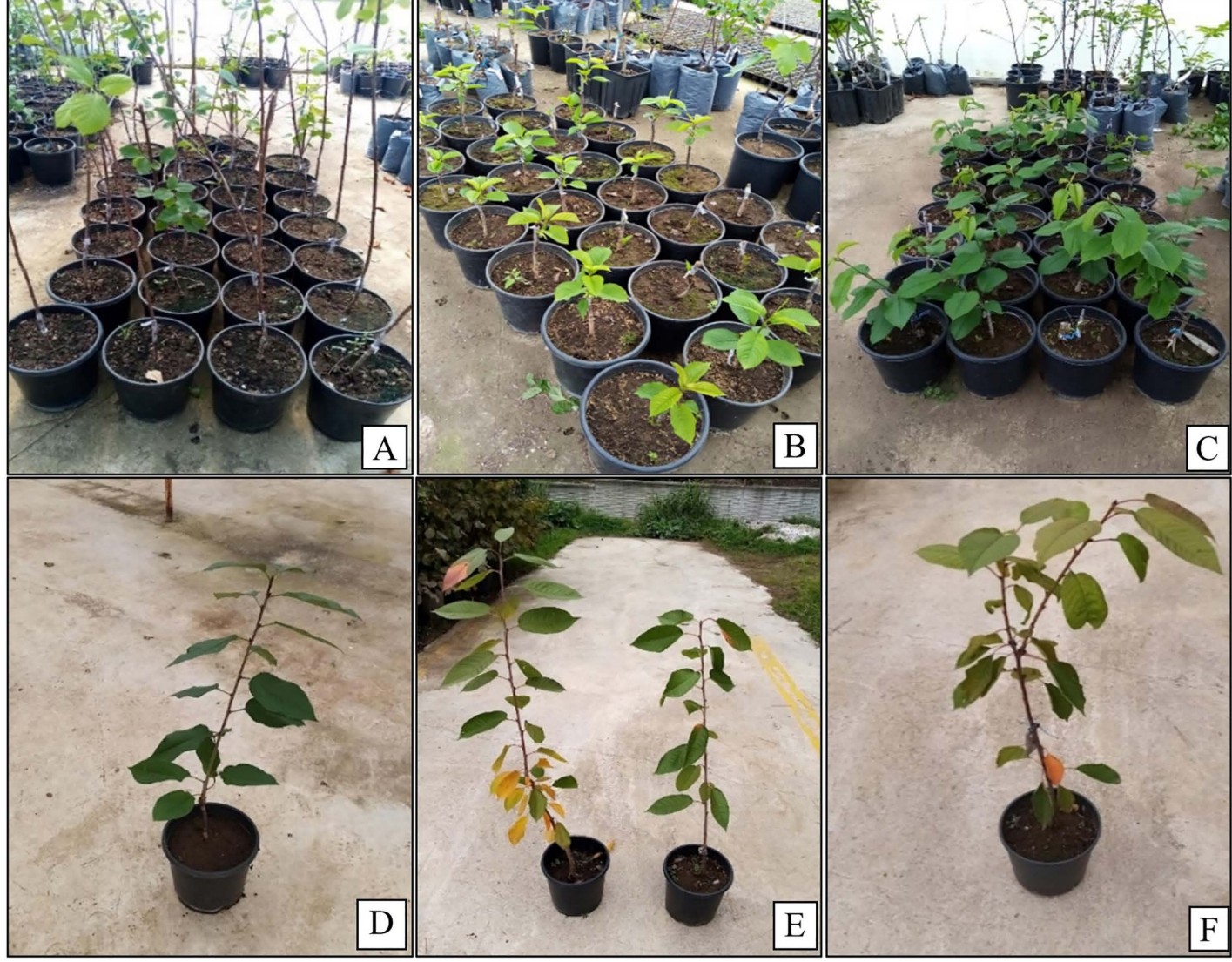

**Fig 3. Shoot development in variety/rootstock combinations . (A and D):08 K 056/0900 Ziraat, (B and E):55 K 104/Lambert, (C and F): 55 V 004/0900 Ziraat.**

growth is a strong positive indicator of compatibility. This result is consistent with physiological expectations, as robust bud development reflects successful vascular integration and signal transmission between scion and rootstock.

Following this, cambium continuity and ability to form callus formation also showed high importance scores. These traits are closely associated with anatomical union and tissue regeneration at the graft site, reinforcing their biological relevance. Their high rankings suggest that structural integration features are central to compatibility assessment.

Moderate importance was assigned to rootstock/scion diameter ratio, length of graft shoot and starch accumulation indicating that while these traits contribute to model predictions, and their effects are comparatively smaller. Interestingly, traits such as necrotic layer formation, cambium differentiation and time of leaf discoloration which are often interpreted as negative signals in classical assessment ranked lower in importance. This suggests that although these traits may indicate

**Table 3. Data of the measurement related to graft incompatibility of the 'Lambert' cultivar grafted onto different rootstock candidates.**

| Characteristic | Gisela 6 | 08 K 056 | 52 K 063 | 55 K 104 | 28 V 001 | 28 V 003 | 55 V 004 | 52 M 003 | 28 M 005 | SL 64 |
|---|---|---|---|---|---|---|---|---|---|---|
| Ability to form callus* | 9.0 | 8.0 | 6.0 | 8.33 | 7.67 | 7.33 | 7.67 | 5.33 | 4.0 | 8.0 |
| Cambium differentiation* | 9.0 | 7.33 | 6.0 | 8.33 | 7.33 | 7.0 | 7.33 | 5.0 | 5.67 | 7.0 |
| Necrotic layer formation* | 10.0 | 8.33 | 6.33 | 7.0 | 7.33 | 7.33 | 7.67 | 6.33 | 4.0 | 8.0 |
| Cambium continuity* | 10.0 | 8.0 | 6.33 | 8.33 | 8.0 | 7.33 | 8.0 | 5.0 | 7.0 | 8.33 |
| Graft bud growth rate (%) | 81.08 | 70.04 | 69.71 | 74.07 | 71.61 | 72.16 | 68.27 | 54.07. | 40.26 | 74.04 |
| Length of graft shoot (cm) | 72.36 | 68.14 | 55.84 | 85.47 | 69.11 | 75.68 | 68.72 | 43.62 | 45.07 | 89.39 |
| Rootstock/scion diameter | 0.92 | 0.84 | 0.66 | 0.68 | 0.71 | 0.75 | 0.77 | 0.56 | 0.47 | 0.81 |
| Defoliation time* | 7.67 | 9.0 | 6.7 | 8.0 | 7.67 | 7.33 | 7.33 | 6.67 | 5.0 | 7.67 |
| Time of leaf discoloration* | 9.0 | 7.67 | 7.0 | 7.33 | 7.33 | 7.0 | 7.0 | 4.0 | 6.0 | 7.33 |
| Starch accumulation* | 10 | 7.67 | 6.0 | 7.0 | 7.33 | 7.0 | 7.33 | 5.0 | 6.0 | 8.0 |

*(1–10) Visual evaluation was made on a 1–10 scale.

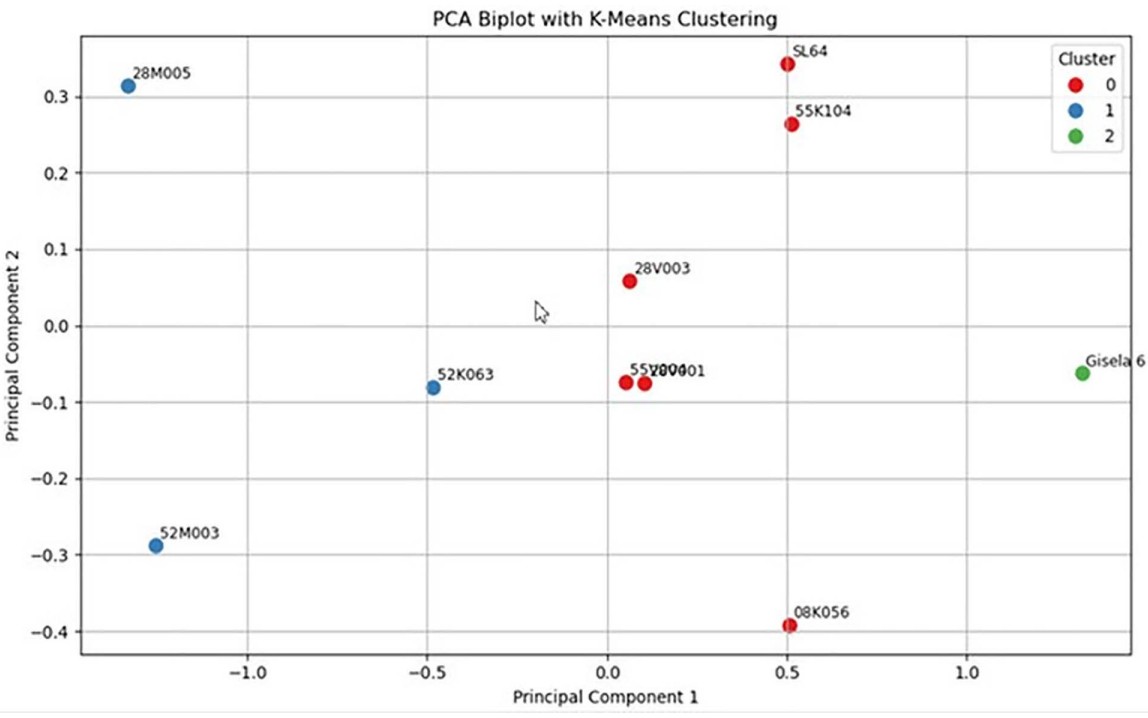

**Fig 4. Two dimensional biplot obtained from Principal Component Analysis (PCA) followed by K-Means clustering based on all normalized trait variables.**

incompatibility, they may not be the most reliable predictors when all traits are considered simultaneously in a multivariate machine learning framework.

The feature importance distribution reinforces the advantage of using ensemble learning methods like Random Forests to uncover hidden nonlinear relationships between traits and graft performance. These insights serve as a precursor to the more interpretable SHAP analysis and validate the inclusion of top-ranked traits in the subsequent Bayesian ranking model.

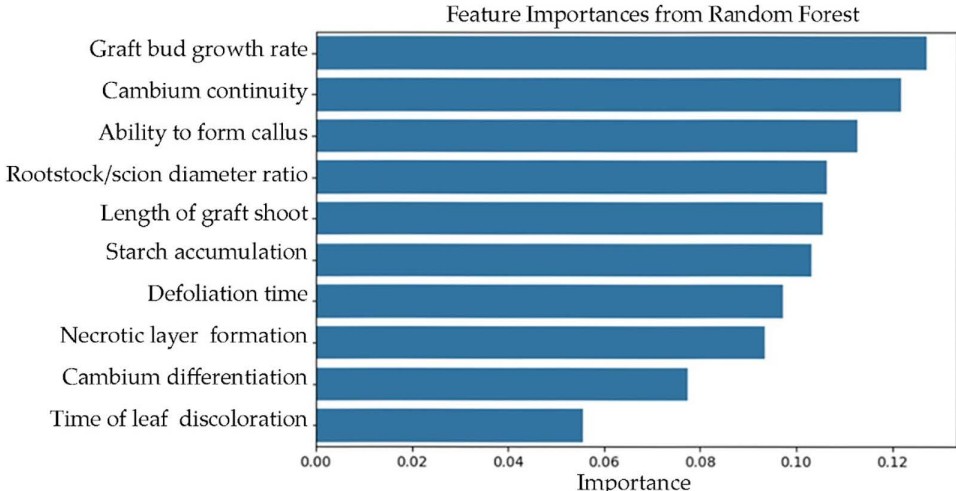

**Fig 5. Feature importance rankings derived from random forest model.**

SHAP summary plot, illustrates the direction and magnitude of each trait's impact on the predicted compatibility score derived from the Random Forest model. Each dot represents a genotype, coloured according to the value of the respective feature (blue = low, red = high). Traits at the top are most influential. SHAP values reflect how much each trait increases (positive value) or decreases (negative value) the predicted score (Fig 6).

The SHAP summary plot presented in Fig 6. provides a detailed and interpretable decomposition of the Random Forest model's predictions for graft compatibility. Unlike traditional feature importance scores, SHAP values quantify the magnitude and direction of each individual feature's contribution to the prediction for each genotype.

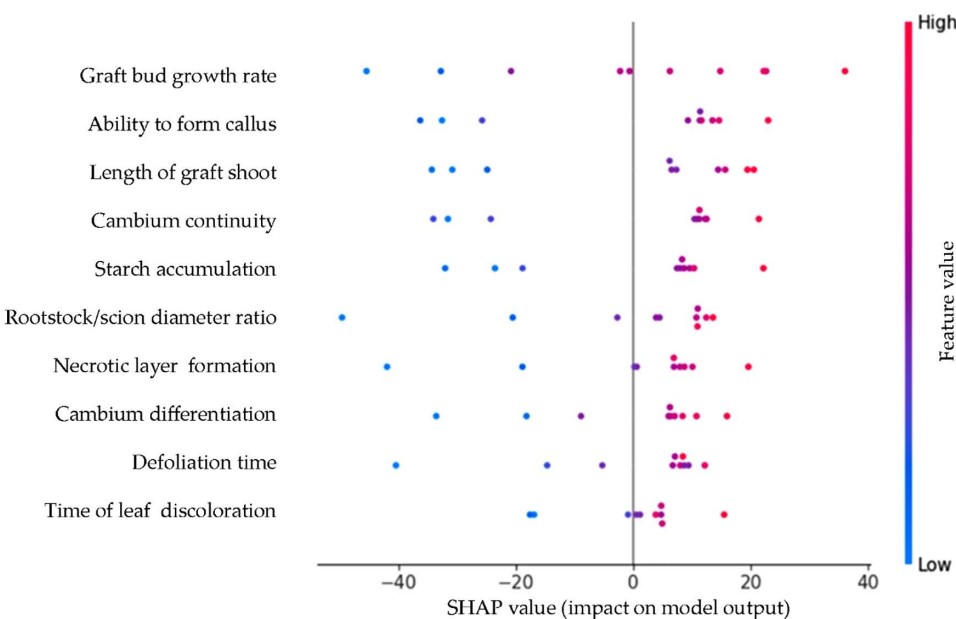

**Fig 6. SHAP summary plot showing trait-wise contribution to compatibility score.**

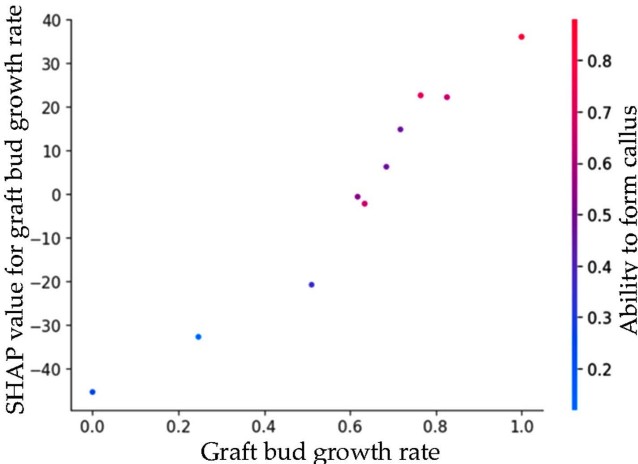

At the top of the plot, graft bud growth rate is confirmed as the most influential variable. High values of this trait (represented in red) consistently shift the prediction to the right, increasing the compatibility score. This confirms the physiological relevance of bud vigor in successful grafting. Similarly, ability to form callus and length of graft shoot appear as strong positive contributors when these traits are high, SHAP values are predominantly positive, reinforcing their favorable role in graft performance.

Cambium continuity and starch accumulation show intermediate levels of impact, and their effects are mostly positive, though with some overlap in low and high values indicating genotype dependent variation. Interestingly necrotic layer formation, cambium differentiation, defoliation time and time of leaf discoloration are associated with negative SHAP values, particularly when their values are high (red). This means that elevated presence of these traits tends to reduce the predicted compatibility score, aligning with their classical interpretation as stress or incompatibility indicators.

Overall, this SHAP analysis validates and extends the feature importance ranking by offering instance level insight. It demonstrates that not only which traits matter, but also how they matter, providing a richer, model agnostic explanation that bridges machine learning with biological interpretability.

SHAP dependence plot in Fig 7. illustrates the relationship between graft bud growth rate values and their corresponding SHAP values (impact on model prediction). Each point represents a genotype. Color intensity indicates the level of ability to form callus with blue denoting low and red denoting high callus values. A strong positive correlation is observed between graft bud growth rate and its SHAP value, suggesting that increases in bud growth consistently raise the predicted compatibility score, especially when callus formation is also high.

SHAP dependence plot for graft bud growth rate, offering a granular view into how this single trait contributes to the model's output and how it interacts with another biologically meaningful variable: ability to form callus. The x-axis represents the observed values of graft bud growth rate, while the y-axis shows the SHAP values for this feature, i.e., its contribution to the final compatibility prediction for each genotype (Fig 7).

A distinct upward trend is visible genotypes with low graft bud growth rate (left-hand side of the graph) have strongly negative SHAP values, indicating a decrease in predicted compatibility. Conversely, as graft bud growth rate increases, its SHAP value rises steeply, ultimately reaching positive contributions exceeding +30 in some cases. This strong monotonic relationship suggests that graft bud growth rate is not only important globally (as shown in Fig 7) but also reliably influences the model's prediction across individual samples.

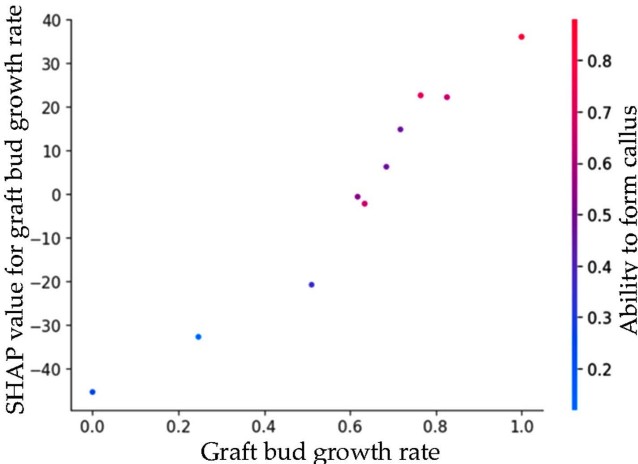

**Fig 7. SHAP dependence plot for bud growth rate, colored by callus formation.**

The color gradient adds a second layer of insight by showing the levels of ability to form callus. Interestingly, genotypes with both high graft bud growth rate and high ability to form callus (upper-right red dots) tend to yield the highest SHAP values. This suggests a potential interaction effect the positive influence of graft bud growth rate on compatibility is amplified when ability to form callus is also abundant. This aligns with the biological rationale that successful grafting relies on both active shoot development and effective tissue regeneration.

Altogether, this dependence plot exemplifies how SHAP can be used to dissect not only the magnitude of feature impact but also the interactions among traits that are not explicitly modelled in the Random Forest algorithm. This level of interpretability strengthens the biological credibility of the model and supports the selection of graft bud growth rate and ability to form callus as key predictors for downstream Bayesian ranking.

Table 4 depicts posterior means, standard deviations (SD), 95% highest density intervals (HDI), Monte Carlo standard errors (mcse), and convergence diagnostics (effective sample size for bulk/tail and Gelman Rubin $\hat{R}$) for the Bayesian regression model estimating the effect of each trait on graft compatibility score. Traits are listed in descending order of conceptual importance. All $\hat{r}$ values indicate convergence ($\leq 1.01$).

This table summarizes the posterior distributions of trait-specific effects estimated using a Bayesian linear regression model. The mean values represent the expected direction and strength of each trait's contribution to compatibility scores across genotypes. Notably, Length of graft shoot (mean = 1.025) and Ability to form callus (mean = 0.993) exhibit the strongest positive effects, with 95% HDI intervals suggesting high certainty about their beneficial role.

Other predictors such as rootstock/scion diameter ratio, starch accumulation, and time of leaf discoloration also show moderate positive effects with narrow HDIs. In contrast, defoliation time (mean = −0.610) and graft bud growth rate (mean = −0.443) have negative means, indicating potentially adverse impacts, though their HDIs include zero, reflecting greater uncertainty.

All convergence diagnostics and high effective sample sizes support the reliability of these posterior estimates. These findings align with the SHAP and Random Forest analyses, reinforcing the importance of shoot vigour, vascular regeneration (callus), and anatomical compatibility in predicting graft success.

Table 5 presents the posterior mean estimates (θ) of each genotype, 95% Highest Density Intervals (HDI), mean ranks across posterior samples, and the probability of being ranked best (Probability Best). The ranking is derived from a Bayesian hierarchical model using MCMC sampling with 4 chains, 1.000 tuning iterations, and 3.000 posterior draws per chain (total 12.000).

**Table 4. Bayesian posterior summary of trait effects on compatibility score.**

| Parameter | mean | sd | hdi_2.5% | hdi_97.5% | mcse_mean | mcse_sd | ess_bulk | ess_tail | $\hat{R}$ |
|---|---|---|---|---|---|---|---|---|---|
| İntercept | −1.919 | 0.246 | −2.387 | −1.431 | 0.005 | 0.003 | 3517 | 2690 | 1 |
| Graft bud growth rate | −0.443 | 0.865 | −2.169 | 1.207 | 0.015 | 0.013 | 3232 | 2419 | 1 |
| Ability to form callus | 0.993 | 0.732 | −0.481 | 2.384 | 0.012 | 0.009 | 3479 | 3524 | 1 |
| Length of graft shoot | 1.025 | 0.598 | −0.185 | 2.14 | 0.009 | 0.006 | 5053 | 4287 | 1 |
| Cambium continuity | −0.187 | 0.806 | −1.781 | 1.36 | 0.013 | 0.01 | 3941 | 3809 | 1 |
| Starch accumulation | 0.838 | 0.76 | −0.678 | 2.294 | 0.011 | 0.008 | 4444 | 4348 | 1 |
| Rootstock/scion diameter ratio | 0.822 | 0.435 | −0.067 | 1.647 | 0.005 | 0.005 | 4373 | 3974 | 1 |
| Necrotic layer formation | 0.342 | 0.612 | −0.877 | 1.577 | 0.008 | 0.007 | 6450 | 4921 | 1 |
| Cambium differentiation | 0.136 | 0.611 | −1.051 | 1.373 | 0.009 | 0.007 | 4680 | 5159 | 1 |
| Defoliation time | −0.610 | 0.644 | −1.825 | 0.696 | 0.006 | 0.006 | 5202 | 4245 | 1 |
| Time of leaf discoloration | 0.591 | 0.783 | −0.216 | 2.141 | 0.012 | 0.009 | 4362 | 4552 | 1 |
| Sigma | 0.239 | 0.132 | 0.069 | 0.498 | 0.004 | 0.003 | 933 | 2256 | 1.01 |

**Table 5. Posterior estimates and ranking probabilities of genotypes based on Bayesian modeling.**

| Cultivars/Genotype | Posterior Mean Theta | HDI 2.5% | HDI 97.5% | Mean Rank | Probability Best |
|---|---|---|---|---|---|
| Gisela 6 | 1.309842 | 0.827155 | 1.73925 | 1.0645 | 0.953917 |
| 55 K 104 | 0.734281 | 0.266295 | 1.13522 | 2.542417 | 0.02675 |
| SL 64 | 0.690912 | 0.267555 | 1.104633 | 2.775583 | 0.017667 |
| 28 V 003 | 0.359229 | 0.019771 | 0.618585 | 4.801333 | 0 |
| 08 K 056 | 0.284924 | −0.125588 | 0.784023 | 5.520417 | 0.001583 |
| 55 V 004 | 0.279234 | −0.082046 | 0.593631 | 5.549917 | 0.000083 |
| 28 V 001 | 0.255202 | −0.046757 | 0.513625 | 5.75475 | 0 |
| 52 K 063 | −0.702571 | −1.02704 | −0.247549 | 8.0825 | 0 |
| 52 M 003 | −1.024151 | −1.468924 | −0.564763 | 8.924417 | 0 |
| 28 M 005 | −2.040137 | −2.465459 | −1.245655 | 9.984167 | 0 |

Gisela 6 rootstock exhibits the highest posterior mean (1.3098), with a narrow and fully positive 95% Highest Density Interval (HDI: [0.8272, 1.7393]), indicating strong evidence of high graft compatibility. It also holds the lowest average rank (Mean Rank = 1.06) and has the highest probability of being the best genotype (Probability Best = 0.9539). These results confirm Gisela 6 rootstock as the most probable optimal rootstock across MCMC simulations. With a posterior mean of 0.7343 and a 95% HDI of [0.2663, 1.1352], 55 K 104 genotype ranks as the second-strongest genotype in the analysis. Although its uncertainty interval is wider than Gisela 6 rootstock, it still lies entirely in the positive domain. Its Mean Rank is 2.54, and the probability of being the top genotype is approximately 2.7%, suggesting it is a competitive but clearly secondary candidate. SL 64 rootstock has a posterior mean of 0.6910 and a similar HDI ([0.2676, 1.1046]) to 55 K 104 genotype. It ranks slightly lower in terms of Mean Rank (2.77) and has a modest probability best of 1.8%. SL 64 rootstock shows consistent but slightly lower graft compatibility than the top two.

28 V 003 genotype shows a posterior mean of 0.3592, and its HDI ranges from 0.0198 to 0.6186. While the entire interval lies in the positive range, the values suggest a moderate effect. Its average rank is 4.80, and its probability of being the best genotype is close to zero, indicating middle tier performance. With a posterior mean of 0.2849 and a 95% HDI of [−0.1266, 0.7782], 08 K 056 genotype demonstrates moderate graft compatibility with notable uncertainty. The inclusion of zero in the interval suggests potential variability. Its Mean Rank is 5.52, and its Probability Best is 0.15%. 55 V 004 genotype posterior mean is 0.2792, with an HDI of [−0.0820, 0.5936]. Although mostly positive, the interval includes negative values, highlighting ambiguity in its performance. It's Mean Rank of 5.55 and Probability Best of 0.0083 reflect its lower likelihood of outperforming others.

The posterior mean of 28 V 001 genotype is 0.2552 with an HDI of [−0.047, 0.5137]. Similar to 55 V 004 genotype, this result is inconclusive, and the genotypic performance is likely inferior to the top candidates. The average rank is 5.75 with zero chance of being the best. 52 K 063 genotype is the first genotype to show an entirely negative 95% HDI [−1.0270, −0.2476] and a posterior mean of −0.703. It's Mean Rank of 8.08 and Probability Best of 0.0 confirm that this genotype consistently underperforms. With a posterior mean of 1.025 and a 95% HDI of [−1.4689, −0.5648], 52 M 003 genotype ranks among the least compatible genotypes. It has a Mean Rank of 8.92 and a Probability Best of essentially 0%, indicating low suitability. 28 M 005 genotype has the lowest posterior mean (−2.04) and the most negative HDI ([−2.465, −1.245]). Its Mean Rank is 9.98, placing it firmly at the bottom. The model assigns it a zero probability of being the best, making it the least promising genotype in the study.

## 4. Discussion

Graft incompatibility, defined as the physiological and anatomical failure at the union of scion and rootstock, represents a major obstacle to the productivity and longevity of *Prunus avium* orchards. Incompatibility often manifests through the formation, thickness, and persistence of necrotic layers at the graft interface, which are considered key anatomical markers

of localized graft failure in cherry rootstock-scion combinations [34,35]. Evaluations based on the status of necrotic layers, callus development, cambial differentiation, and new vascular tissue formation have been shown to reliably indicate the compatibility of grafted combinations across various fruit species [36,37].

As shown in significant differences were observed among the graft combinations in terms of callus formation ability, with the Gisela 6 and SL 64 rootstocks, as well as the 08 K 056 and 55 K 104 genotypes, exhibiting the highest values. The observed differences in callus formation ability are consistent with previous studies that emphasize the importance of callus development in determining graft incompatibility (Tables 2 and 3). For instance, in apricot grafted onto Pixy rootstock, weak callus formation was observed 15 days post-grafting, whereas strong cambial activity and vascular differentiation were present in compatible combinations [36].

As presented in (Tables 2 and 3), the genotypes (08 K 056, 55 K 104, 28 V 001, 28 V 003), as well as the rootstocks Gisela 6 and SL 64, exhibited high levels of cambial continuity. These findings support the hypothesis that the integrity of vascular connections is a critical factor influencing graft compatibility and success. Additionally, with respect to necrotic layer formation, the genotypes (52 K 063, 28 M 005 and 52 M 003) exhibiting lower values provide further evidence of graft incompatibility, which may be attributed to underlying biochemical incompatibilities between scion and rootstock tissues. Similarly, in anatomical studies involving '0900 Ziraat' and 'Starks Gold' cultivars grafted onto Gisela 5 and SL 64 rootstock, a necrotic surface and limited cambial continuity were reported at six months, yet by twelve months, successful vascular integration had occurred [37]. In another study, callus development began two weeks after grafting in sweet cherry combinations involving various hybrid rootstocks [38].

The vascular structure and water transport capacity of the rootstock have also been identified as critical factors influencing compatibility. For example, in incompatible peach/plum grafts, most callus cells failed to differentiate, and vascular development remained incomplete after four months, accompanied by increased necrosis [39]. Other investigations across fruit species showed that, by 120 days after grafting, callus tissues exhibited parenchymatic characteristics, cambial activity persisted, and vascular connections (xylem and phloem) were established, although necrotic layers, while still present, had diminished compared to earlier stages [36,38,40].

Anatomical analyses reported in sources [41–44] evaluated graft compatibility among different *Prunus* rootstocks and demonstrated successful graft union after twelve months. Distinct structural indicators supporting anatomical compatibility were identified at the graft interface, including cambial continuity, vascular differentiation, organized parenchymatic tissue, and occasional sclereid formation.

In the present study, differences in callus formation, cambial differentiation, necrotic layer development, and cambial continuity among rootstock candidate genotypes are likely attributed to factors such as grafting technique, timing, and the specific scion rootstock species involved. Consistent with our findings, previous research has also highlighted the influence of genetic diversity on bud break and graft success [45–47]. Rootstock scion incompatibility remains a welldocumented challenge in cherry cultivation, with significant impacts on graft bud growth and overall plant performance [47,48]. The study evaluating graft incompatibility between apricot cultivars and different *Prunus* rootstocks revealed a significant positive association between graft take and graft length [49]. Many studies have shown differences in the length of the graft shoots among different cherry genotypes and rootstocks are probably due to related genetic differences as stated by [47,49,50]. Low rootstock/scion diameter values may be related to abnormalities in graft union resulting from blockages in assimilates and water transport [51]. In accordance with our study results, stated that genetic diversity has an effect on length of graft shoot and rootstock/scion diameter. The other symptoms of incompatible graft combinations include yellow and red leaves, leaf curling, and reddening of shoots [47]. Leaf chlorosis and abscission occur over time due to decreased carbohydrate translocation at the scion and rootstock growth junction [52,53]. Low accumulation of starch at the top and bottom of the graft causes better boiling of the graft and adequate transfer of nutrients to the roots [54]. Changes in starch levels, especially the accumulation above the graft union and its absence below it, can lead to damage in phloem vessels. This accumulation has been reported to affect the overall health of the plant by inhibiting nutrient transport [55,56].

Beyond anatomical indicators, statistical and machine learning analyses in this study provided robust insights into trait contributions and genotype performance. Principal Component Analysis (PCA) followed by K-Means clustering revealed clear groupings among genotypes based on multi-trait similarity, highlighting patterns of compatibility and performance under complex trait interactions. The Random Forest model, trained on normalized trait data, achieved high predictive accuracy, and SHAP analysis revealed that necrotic layer formation had a strong negative contribution to graft incompatibility predictions. This finding is consistent with known physiological barriers to successful graft union formation. Excessive necrosis at the graft site is known to impede callus bridge formation and prevent the reconnection of xylem and phloem tissues. Such disconnection obstructs the transport of carbohydrates, water, and signalling molecules, ultimately leading to graft failure. This mechanistic insight reinforces the predictive utility of SHAP while also providing a biologically interpretable explanation for its outputs (Fig 6).

Moreover, Bayesian ranking models offered probabilistic evaluations of genotype performance, where Gisela 6 consistently ranked highest with narrow credible intervals, affirming its known compatibility and vigour. Among candidate genotypes, 08 K 056 showed promising compatibility with both cultivars, as evidenced by anatomical scores and high posterior probabilities in the Bayesian framework. In contrast, 28 M 005 and 52 M 003 exhibited low callus formation and continuity, frequent necrosis, and inferior statistical rankings, reinforcing their limited suitability as rootstocks.

In recent years, machine-learning approaches have gained momentum in horticultural research, particularly in modelling complex fruit traits and physiological responses in grafted plants. These methods have shown potential for predicting fruit weight, anatomical compatibility, and even textural properties using non-destructive measurements. However, most previous applications have focused primarily on achieving high prediction accuracy, often lacking in model interpretability and biological transparency. In contrast, the present study introduces an explainable and reproducible hybrid framework that combines statistical and machine learning techniques to not only predict graft compatibility but also identify the most influential anatomical and physiological traits. This integrative perspective enhances our understanding of trait-level interactions and provides valuable insights for breeding programs aiming to improve grafting outcomes in sweet cherry cultivars.

Overall, these data-driven insights reinforce the role of both anatomical and biochemical factors in graft incompatibility, and demonstrate the utility of multi-dimensional evaluation approaches in rootstock selection. The integration of machine learning and Bayesian inference not only enhances interpretability and objectivity but also facilitates data-driven rootstock selection in breeding programs.

## 5. Conclusions

This study evaluated the graft compatibility of five cherry rootstock candidates (three sour cherry and two sweet cherry genotypes) against standard rootstocks Gisela 6 and SL 64 using '0900 Ziraat' and 'Lambert' sweet cherry cultivars as scions. Morphological, anatomical, and physiological parameters were assessed to reveal compatibility patterns. Gisela 6 consistently demonstrated superior performance across most metrics, serving as a benchmark for evaluating the candidate genotypes.

To enhance the robustness and interpretability of the findings, data-driven approaches such as Principal Component Analysis (PCA), Random Forest with SHAP values, and Bayesian ranking were employed. PCA and k-means clustering revealed clear differentiation among genotypes, indicating inherent trait-based groupings. Random Forest modeling identified key traits contributing to compatibility such as callus formation, bud burst rate, and starch accumulation while SHAP values allowed for transparent interpretation of individual genotype performance. Furthermore, Bayesian analysis enabled probabilistic ranking of genotypes, highlighting 28 M 005 and 52 M 003 as the least compatible, whereas Gisela 6 and SL 64 maintained the highest posterior compatibility scores.

These integrative results suggest that while some local genotypes show potential, their compatibility with widely cultivated sweet cherry varieties remains limited under current conditions. Therefore, combining classical evaluation with

machine learning-based analyses offers a robust framework for future rootstock selection and breeding programs aimed at enhancing sweet cherry orchard sustainability.

Although our findings offer valuable insights into early-stage graft compatibility, it is important to recognize that long-term agronomic traits such as yield, fruit quality, and stress resilience were not evaluated in this study. Future research should focus on validating the model's robustness across multiple environments and over extended periods, which would significantly enhance the translational potential and practical utility of the proposed machine learning framework.

## Author contributions

**Data curation:** Mehmet Ali Cengiz.

**Formal analysis:** Mehmet Ali Cengiz.

**Investigation:** Erol Aydin, Ercan Er, Hüsnü Demirsoy.

**Methodology:** Erol Aydin, Mehmet Ali Cengiz, Hüsnü Demirsoy.

**Resources:** Ercan Er.

**Writing – original draft:** Erol Aydin.

**Writing – review & editing:** Hüsnü Demirsoy.

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
