## [Decision Letter · Decision Letter 0]

20 Aug 2025

PONE-D-25-42641Identifying Graft Incompatible Rootstocks for Sweet Cherry Through Machine Learning AlgorithmsPLOS ONE

Dear Dr. AYDIN,

Thank you for submitting your manuscript to PLOS ONE. After careful consideration, we feel that it has merit but does not fully meet PLOS ONE’s publication criteria as it currently stands. Therefore, we invite you to submit a revised version of the manuscript that addresses the points raised during the review process.

We look forward to receiving your revised manuscript.

Kind regards,

Yuan Huang

Academic Editor

PLOS ONE

Journal Requirements:

3. Please note that PLOS One has specific guidelines on code sharing for submissions in which author-generated code underpins the findings in the manuscript. In these cases, we expect all author-generated code to be made available without restrictions upon publication of the work. Please review our guidelines at https://journals.plos.org/plosone/s/materials-and-software-sharing#loc-sharing-code and ensure that your code is shared in a way that follows best practice and facilitates reproducibility and reuse.

“This work were funded by the Deanship of Scientific Research at Imam Mohammad Ibn Saud Islamic University (IMSIU) (grant number IMSIU-DDRSP2502).”

6. We note that your Data Availability Statement is currently as follows: All relevant data are within the manuscript and its Supporting Information files.

7. When completing the data availability statement of the submission form, you indicated that you will make your data available on acceptance. We strongly recommend all authors decide on a data sharing plan before acceptance, as the process can be lengthy and hold up publication timelines. Please note that, though access restrictions are acceptable now, your entire data will need to be made freely accessible if your manuscript is accepted for publication. This policy applies to all data except where public deposition would breach compliance with the protocol approved by your research ethics board. If you are unable to adhere to our open data policy, please kindly revise your statement to explain your reasoning and we will seek the editor's input on an exemption. Please be assured that, once you have provided your new statement, the assessment of your exemption will not hold up the peer review process.

8. PLOS requires an ORCID iD for the corresponding author in Editorial Manager on papers submitted after December 6th, 2016. Please ensure that you have an ORCID iD and that it is validated in Editorial Manager. To do this, go to ‘Update my Information’ (in the upper left-hand corner of the main menu), and click on the Fetch/Validate link next to the ORCID field. This will take you to the ORCID site and allow you to create a new iD or authenticate a pre-existing iD in Editorial Manager.

9. We note that Figure 1 in your submission contain [map/satellite] images which may be copyrighted. All PLOS content is published under the Creative Commons Attribution License (CC BY 4.0), which means that the manuscript, images, and Supporting Information files will be freely available online, and any third party is permitted to access, download, copy, distribute, and use these materials in any way, even commercially, with proper attribution. For these reasons, we cannot publish previously copyrighted maps or satellite images created using proprietary data, such as Google software (Google Maps, Street View, and Earth). For more information, see our copyright guidelines: http://journals.plos.org/plosone/s/licenses-and-copyright.

**Additional Editor Comments:**

Please consider the comments from two reviewers.

Reviewers' comments:

Reviewer's Responses to Questions

**Comments to the Author**

1. Is the manuscript technically sound, and do the data support the conclusions?

Reviewer #1: Yes

Reviewer #2: Yes

2. Has the statistical analysis been performed appropriately and rigorously? 

Reviewer #1: Yes

Reviewer #2: Yes

3. Have the authors made all data underlying the findings in their manuscript fully available?

Reviewer #1: Yes

Reviewer #2: Yes

4. Is the manuscript presented in an intelligible fashion and written in standard English?

Reviewer #1: Yes

Reviewer #2: Yes

5. Review Comments to the Author

Reviewer #1: This study evaluated the graft incompatibility by identifying Graft Incompatible Rootstocks for Sweet Cherry Through Machine Learning Algorithms.It is meaningful to study the topic, and the whole story is complete.However, minor revisions are necessary to increase the readability and rigor of the article.

Comments are as follows,

1.Problems in the abstract:The author has made a lot of descriptions of the research method, but only one sentence describes the result analysis, and the result analysis part should be emphasized.

2.For Figures 3 and 4, please explain why all grafted combinations are not shown? It is recommended to supplement the phenotypic picture of all grafted assemblies completely.

3.In the discussion part, it is recommended to discuss the data of this study, and it is recommended to cite the corresponding chart when discussing the content of your own research.

Reviewer #2: This study employs machine learning (ML) techniques to evaluate graft compatibility in cherry rootstocks. Three ML algorithms—Principal Component Analysis (PCA) with clustering, Random Forest (RF) with SHAP interpretation, and Bayesian ranking—were applied to analyze phenotypic data from graft unions, identifying key compatibility traits and probabilistically ranking genotype performance. The research identified five high-compatibility candidate genotypes (e.g., 55 K 104) and validated the superiority of Gisela 6. While manual evaluation of 12-month phenotypic data can assess basic compatibility, ML provides unparalleled capabilities in quantifying multi-trait interactions, revealing hidden biological mechanisms, and enabling risk-informed breeding decisions. Although the study shows significant potential, several critical issues require resolution. Major revision is recommended to address the following points:

1.Although precedents exist for applying machine learning in plant science, the paper fails to adequately contextualize its work against these prior studies and does not clarify the unique contributions of this research. For instance, the specific novelty of using SHAP and Bayesian ranking in the context of graft compatibility is not thoroughly discussed; it is only vaguely mentioned as 'overcoming the limitations of traditional statistics.' It is recommended that the authors add comparisons with similar studies (e.g., machine learning applications in apricot grafting, Reference 45) in the Introduction or Discussion section to highlight the advantages of their approach."

2.The study focuses on short-term compatibility but does not explore long-term agronomic traits (e.g., yield, stress resilience), which limits its translational value. It is recommended that the Discussion or Conclusion section emphasize the need for future multi-environment trials to validate the robustness of machine learning predictions, thereby enhancing practical applicability.

3. In the Methods section (2.2), the analysis mentions "n=5" but does not clarify whether this refers to sub-samples per genotype-scion combination or the total sample. This ambiguity may compromise reproducibility.

4. Hyperparameters for the Random Forest and Bayesian models are not detailed. For example, the Bayesian model uses "4 chains, 1,000 tuning iterations" but omits convergence criteria (e.g., Gelman-Rubin statistic). It is recommended to add a subsection (e.g., "2.2.3 Machine Learning Parameters") listing key parameters to ensure reproducibility.

5. Figure 3 illustrates graft development but is only briefly mentioned in the Results (Section 3.1), with no linkage to quantitative data (e.g., graft bud growth rate). It is advised to directly reference Figure 3 in the Results to strengthen visual evidence.

6. K-means clustering did not validate the cluster number selection, potentially leading to subjective grouping (Figure 5). It is recommended to supplement clustering with validation metrics (e.g., elbow method) or acknowledge this limitation in the Discussion.

7. The SHAP summary plot (Figure 7) shows a negative impact of "necrotic layer formation," but the Discussion fails to explain its physiological implications. It is suggested to add a paragraph correlating SHAP outputs (Figures 7–8) with known mechanisms (e.g., carbohydrate transport barriers, Reference 48).

6. PLOS authors have the option to publish the peer review history of their article (what does this mean? ). If published, this will include your full peer review and any attached files.

**Do you want your identity to be public for this peer review?** For information about this choice, including consent withdrawal, please see our Privacy Policy .

Reviewer #1: No

Reviewer #2: No

---

## [Author Response · Author response to Decision Letter 1]

31 Aug 2025

Response to Journal Requirements:

1. Please ensure that your manuscript meets PLOS ONE's style requirements, including those for file naming. The PLOS ONE style templates can be found at:

We sincerely thank you for your editorial feedback and the opportunity to revise our manuscript. In response to your request regarding adherence to PLOS ONE’s formatting and submission guidelines, we confirm that the following actions have been completed:

1. Formatting Compliance: The manuscript has been thoroughly revised to comply with the formatting requirements outlined in the following documents:

*PLOS ONE formatting sample main body

*PLOS ONE formatting sample title, authors, affiliations

2. File Naming: All uploaded files, including the main manuscript, figures, and supplementary materials, have been renamed in accordance with PLOS ONE’s file naming conventions (e.g., “Main Manuscript.docx”, “Figure1.tif”, “SupportingInfo_S1.xlsx”).

3. Style and Structure: The manuscript structure (including Abstract, Introduction, Materials and Methods, Results, Discussion, References) now follows the journal’s required order and formatting. Headings have been adjusted for consistency, and line numbering has been applied to facilitate further review.

We hope that the revised submission meets all journal requirements. Please do not hesitate to let us know if any further clarification or adjustment is needed.

We thank the editor for the helpful clarification request regarding field site permissions. In response, we have added the following sentence at the beginning of 2.2. Methods section:

“All experimental data were collected from trees grown at the Black Sea Agricultural Research Institute (Gelemen, Samsun, Turkey) with prior permission granted by the Institute’s administration.”

This addition ensures that the manuscript complies with PLOS ONE’s requirements concerning field site authorization and clearly indicates the institutional approval obtained for this research.

3.Please note that PLOS One has specific guidelines on code sharing for submissions in which author-generated code underpins the findings in the manuscript. In these cases, we expect all author-generated code to be made available without restrictions upon publication of the work. Please review our guidelines at [….] and ensure that your code is shared in a way that follows best practice and facilitates reproducibility and reuse.

We thank the editor for the reminder regarding PLOS ONE’s code sharing policy. In compliance with these guidelines, all Python code used in this study has been made publicly available via Zenodo, ensuring full transparency and reproducibility.

The repository can be accessed at the following DOI:

https://doi.org/10.5281/zenodo.16944149b

This link has been:

• Cited within the manuscript and

• Included in the cover letter

We confirm that the shared code allows readers to reproduce the computational procedures described in the paper, in line with best practices for open science.

4.We note that the grant information you provided in the ‘Funding Information’ and ‘Financial Disclosure’ sections do not match. When you resubmit, please ensure that you provide the correct grant numbers for the awards you received for your study in the ‘Funding Information’ section.

We thank the editor for pointing out the discrepancy between the Funding Information and Financial Disclosure sections. In response, we have carefully revised both sections to ensure consistency.

The correct grant number, IMSIU-DDRSP2502, is now listed identically in both sections.

This update ensures alignment with PLOS ONE’s formatting standards for funding disclosures.

5. Please state what role the funders took in the study. If the funders had no role, please include the following statement:

Also, please include this amended Role of Funder statement in your cover letter.

We thank the editor for this clarification request.

In response, we confirm that:

This statement has been explicitly added to the manuscript and also included in the cover letter as instructed.

6.We note that your Data Availability Statement is currently as follows:

All relevant data are within the manuscript and its Supporting Information files.

Please confirm at this time whether or not your submission contains all raw data required to replicate the results of your study. Authors must share the “minimal data set” for their submission. PLOS defines the minimal data set to consist of the data required to replicate all study findings reported in the article, as well as related metadata and methods

(https://journals.plos.org/plosone/s/data-availability#loc-minimal-data-set-definition

We thank the editor for emphasizing the importance of data transparency.

In response, we have revised the Data Availability Statement to explicitly include the public repository and DOI where the dataset can be accessed. The updated statement now reads:

“The datasets used and analyzed during the current study are publicly available on Zenodo at the following DOI: https://doi.org/10.5281/zenodo.16944180.”

This ensures full compliance with PLOS ONE’s data sharing policy and supports reproducibility and open science practices.

7. When completing the data availability statement of the submission form, you indicated that you will make your data available on acceptance. We strongly recommend all authors decide on a data sharing plan before acceptance, as the process can be lengthy and hold up publication timelines. Please note that, though access restrictions are acceptable now, your entire data will need to be made freely accessible if your manuscript is accepted for publication. This policy applies to all data except where public deposition would breach compliance with the protocol approved by your research ethics board. If you are unable to adhere to our open data policy, please kindly revise your statement to explain your reasoning and we will seek the editor's input on an exemption. Please be assured that, once you have provided your new statement, the assessment of your exemption will not hold up the peer review process

We thank the editor for the detailed clarification regarding the journal’s data availability policy.

We are pleased to confirm that the full dataset used in this study has been made publicly available without restriction via Zenodo. This ensures full compliance with PLOS ONE’s open data policy.

The dataset can be accessed at the following DOI: https://doi.org/10.5281/zenodo.16944180

This updated information has also been included in the revised Data Availability Statement section of the manuscript and in the submission form.

8. PLOS requires an ORCID iD for the corresponding author in Editorial Manager on papers submitted after December 6th, 2016. Please ensure that you have an ORCID iD and that it is validated in Editorial Manager. To do this, go to ‘Update my Information’ (in the upper left-hand corner of the main menu), and click on the Fetch/Validate link next to the ORCID field. This will take you to the ORCID site and allow you to create a new iD or authenticate a pre-existing iD in Editorial Manager.

We appreciate the reminder regarding the ORCID iD requirement for the corresponding author.

We confirm that the corresponding author has a valid ORCID iD, and it has been successfully authenticated and validated within the PLOS Editorial Manager system.

9. We note that Figure 1 in your submission contain [map/satellite] images which may be copyrighted. All PLOS content is published under the Creative Commons Attribution License (CC BY 4.0), which means that the manuscript, images, and Supporting Information files will be freely available online, and any third party is permitted to access, download, copy, distribute, and use these materials in any way, even commercially, with proper attribution. For these reasons, we cannot publish previously copyrighted maps or satellite images created using proprietary data, such as Google software (Google Maps, Street View, and Earth). For more information, see our copyright guidelines: http://journals.plos.org/plosone/s/licenses-and-copyright.

We require you to either (1) present written permission from the copyright holder to publish these figures specifically under the CC BY 4.0 license, or (2) remove the figures from your submission.

Thank you for your note regarding the copyright policy and the use of satellite or map-based images.

In response to your comment, we have removed Figure 1 from the revised manuscript to fully comply with PLOS ONE's licensing and copyright requirements.

We have carefully reviewed the reviewer comments, and no recommendations were made to cite specific previously published works. Therefore, no additional citations were required.

Response to Reviewer #1

1.Problems in the abstract: The author has made a lot of descriptions of the research method, but only one sentence describes the result analysis, and the result analysis part should be emphasized.

We sincerely thank the reviewer for the insightful comment regarding the abstract. In response, we have revised the abstract to provide a clearer and more detailed emphasis on the result analysis. Specifically, we added exact correlation coefficients between key traits (e.g., graft bud growth rate, shoot length and rootstock/scion diameter ratio highlighted the most influential predictors identified by SHAP analysis (such as graft bud growth rate and ability to form callus), and emphasized how these findings contribute to breeding strategies. We also retained a concise explanation of the methodological pipeline to maintain clarity and scientific flow. This revision ensures that the abstract now reflects a balanced summary of the methods and the major findings, thus enhancing the clarity and relevance of the study.

2. For Figures 3 and 4, please explain why all grafted combinations are not shown? It is recommended to supplement the phenotypic picture of all grafted assemblies completely.

We appreciate the reviewer’s comment regarding the completeness of Figures 2 and 3. The current phenotypic images were selected to represent the most characteristic examples of graft compatibility and incompatibility observed across combinations. Unfortunately, due to limitations in image quality and consistency for some grafted assemblies, not all combinations could be included in the figures. However, we ensured that the presented images cover the full spectrum of observed phenotypes and provide a representative overview of the key graft responses. We have now added a clarifying sentence in the figure captions and the Methods section to explain this rationale.

3.In the discussion part, it is recommended to discuss the data of this study, and it is recommended to cite the corresponding chart when discussing the content of your own research.

We thank the reviewer for this valuable suggestion. In the revised manuscript, we have revised the Discussion section to include more detailed commentary on the key findings of this study. We have also added explicit references to the relevant figures and tables when interpreting the results, in order to improve clarity and strengthen the link between the discussion and the presented data.

Response to Reviewer #2

1.Although precedents exist for applying machine learning in plant science, the paper fails to adequately contextualize its work against these prior studies and does not clarify the unique contributions of this research. For instance, the specific novelty of using SHAP and Bayesian ranking in the context of graft compatibility is not thoroughly discussed; it is only vaguely mentioned as 'overcoming the limitations of traditional statistics.' It is recommended that the authors add comparisons with similar studies (e.g., machine learning applications in apricot grafting, Reference 45) in the Introduction or Discussion section to highlight the advantages of their approach.

Thank you for this valuable observation regarding the need to better contextualize our approach in relation to prior studies and to clarify the specific novelty of using SHAP values and Bayesian ranking in the context of graft compatibility.

In response to your suggestion, we have substantially revised the Introduction section to incorporate recent studies that apply machine learning techniques in the domain of grafting and fruit trait prediction. Specifically, we discuss works that utilize deep neural networks, computer vision, and classical machine learning methods for graft compatibility assessment and trait prediction in species such as Camellia oleifera, grapevine, apricot, and plum.

Furthermore, we have clarified the unique contribution of our study in both the Introduction and the Discussion sections. In particular, we now emphasize that, unlike most previous studies that focus primarily on predictive accuracy, our work presents an interpretable hybrid framework combining PCA for dimensionality reduction, Random Forests for predictive modeling, SHAP for explainability, and Bayesian ranking for trait-level prioritization. This integrative framework provides both prediction and insight, helping identify the anatomical and physiological features most relevant to graft compatibility in sweet cherries.

We hope that these revisions sufficiently address your concerns and better highlight the novelty and relevance of our approach within the current literature.

2. "The study focuses on short-term compatibility but does not explore long-term agronomic traits (e.g., yield, stress resilience), which limits its translational value. It is recommended that the Discussion or Conclusion section emphasize the need for future multi-environment trials to validate the robustness of machine learning predictions, thereby enhancing practical applicability."

We appreciate the reviewer’s valuable comment regarding the importance of evaluating long-term agronomic traits. In response, we have expanded the Conclusion section to emphasize this limitation. We now explicitly acknowledge that our study focuses on early-stage graft compatibility, and we have added a statement recommending future multi-environment and long-term trials to assess the robustness and practical applicability of the machine learning predictions. This addition reinforces the translational value of our findings and aligns with the reviewer’s insightful suggestion.

3. "In the Methods section (2.2), the analysis mentions 'n=5' but does not clarify whether this refers to sub-samples per genotype-scion combination or the total sample. This ambiguity may compromise reproducibility."

Thank you for pointing out the ambiguity regarding the sample size. We have now clarified in Section 2.2 that “n=5” refers to five biological replicates per genotype-scion combination. Although data were collected from each replicate individually, the statistical analyses were performed on the average values of these five replicates. This clarification has been added to improve reproducibility and transparency.

4. "Hyperparameters for the Random Forest and Bayesian models are not detailed. For example, the Bayesian model uses '4 chains, 1,000 tuning iterations' but omits convergence criteria (e.g., Gelman-Rubin statistic). It is recommended to add a

---

## [Decision Letter · Decision Letter 1]

7 Sep 2025

Identifying Graft Incompatible Rootstocks for Sweet Cherry Through Machine Learning Algorithms

PONE-D-25-42641R1

Dear Dr. AYDIN,

We’re pleased to inform you that your manuscript has been judged scientifically suitable for publication and will be formally accepted for publication once it meets all outstanding technical requirements.

Kind regards,

Yuan Huang

Academic Editor

PLOS ONE

Additional Editor Comments (optional):

Reviewer #1:

Reviewer #2:

Reviewers' comments:

Reviewer's Responses to Questions

**Comments to the Author**

1. If the authors have adequately addressed your comments raised in a previous round of review and you feel that this manuscript is now acceptable for publication, you may indicate that here to bypass the “Comments to the Author” section, enter your conflict of interest statement in the “Confidential to Editor” section, and submit your "Accept" recommendation.

Reviewer #1: All comments have been addressed

Reviewer #2: All comments have been addressed

2. Is the manuscript technically sound, and do the data support the conclusions?

Reviewer #1: Yes

Reviewer #2: Yes

3. Has the statistical analysis been performed appropriately and rigorously? 

Reviewer #1: Yes

Reviewer #2: Yes

4. Have the authors made all data underlying the findings in their manuscript fully available?

Reviewer #1: Yes

Reviewer #2: Yes

5. Is the manuscript presented in an intelligible fashion and written in standard English?

Reviewer #1: Yes

Reviewer #2: Yes

6. Review Comments to the Author

Reviewer #1: (No Response)

Reviewer #2: The authors have adequately addressed my comments, with the remaining issues pertaining to minor terminological inconsistencies. The enhancement in explicating the novelty of the machine learning framework and the improvement in reproducibility are substantial, meeting the journal's publication standards. It is recommended that the editorial office assist in refining terminology during the typesetting phase prior to acceptable for publication.

7. PLOS authors have the option to publish the peer review history of their article (what does this mean? ). If published, this will include your full peer review and any attached files.

**Do you want your identity to be public for this peer review?** For information about this choice, including consent withdrawal, please see our Privacy Policy .

Reviewer #1: No

Reviewer #2: No

---

## [Editor Report · Acceptance letter]

PONE-D-25-42641R1

PLOS ONE

Dear Dr. AYDIN,

I'm pleased to inform you that your manuscript has been deemed suitable for publication in PLOS ONE. Congratulations! Your manuscript is now being handed over to our production team.

Kind regards,

on behalf of

Dr. Yuan Huang

Academic Editor

PLOS ONE